# Evidence accumulation occurs locally in the parietal cortex

Zhewei Zhang ⓘ [1,2], Chaoqun Yin ⓘ [1] & Tianming Yang ⓘ [1] ✉

Decision making often entails evidence accumulation, a process that is represented by neural activities in a network of multiple brain areas. Yet, it has not been identified where exactly the accumulation originates. We reason that a candidate brain area should both represent evidence accumulation and information that is used to compute evidence. Therefore, we designed a two-stage probabilistic reasoning task in which the evidence for accumulation had to be first determined from sensory signals orthogonal to decisions. With a linear encoding model, we decomposed the responses of posterior parietal neurons to each stimulus into an early and a late component that represented two dissociable stages of decision making. The former reflected the transformation from sensory inputs to accumulable evidence, and the latter reflected the accumulation of evidence and the formation of decisions. The presence of both computational stages indicates that evidence accumulation signal in the parietal cortex is computed locally.

Ramping activities associated with evidence accumulation during decision making are considered as a signature of brain areas involved in decision making. The lateral intraparietal area (LIP) was among the first where neurons were demonstrated to exhibit such an activity pattern and therefore proposed as a critical decision-making area where neurons accumulate evidence for decisions[1,2]. Since then, similar evidence-dependent ramping activities have been demonstrated in many other brain structures, including the prefrontal cortex[3–5], the striatum[6], and the superior colliculus[7–9]. These results, although by themselves not against LIP's role in decision making, raised the question of whether evidence accumulation signals in LIP only reflect the computation carried out somewhere else in the brain. This question cannot be examined with lesions experiments, which have been used to provide clues on whether LIP activities causally contribute to behavior[10–12]. Whether LIP plays a causal role in decision-making or not does not exclude the possibility that LIP sits downstream of where decisions are computed and inherits the evidence accumulation signal from there.

Here, we approached the question from a different angle. We reason that if LIP inherits evidence accumulation signal from somewhere else, we are unlikely to observe computation stages that happen before the accumulation. In particular, the transformation from sensory inputs to accumulable evidence, which has not been well studied in the field, is a necessary computation stage in certain types of decision making. This computation may be trivial in some perceptual decision-making tasks, including the classic random dot motion discrimination task, as the evidence is approximately linearly encoded by the firing rate of the sensory neurons and can be directly accumulated[13,14]. Yet, in other tasks where evidence is conferred via arbitrary stimulus-evidence associations, extra steps are necessary to compute evidence from the sensory information[15–17]. In these cases, we distinguish the evidence that can be readily accumulated based on its neural representation from the raw sensory information that requires additional preprocessing. Although it has been shown that LIP activities represent evidence accumulation in these tasks[15,16], it is unclear where the extra steps take place locally or in a different brain structure. Representations of the sensory-to-evidence transformation in LIP, if demonstrated, would argue strongly for the hypothesis that evidence accumulation is computed in LIP.

To this end, we adapted a probabilistic reasoning task that was previously used to investigate evidence accumulation in the LIP[16]. Just as in the original task, the monkeys were required to accumulate evidence from a series of visual stimuli to form eye movement decisions between a green and a red peripheral target.

[1]Institute of Neuroscience, Key Laboratory of Primate Neurobiology, Center for Excellence in Brain Science and Intelligence Technology, Chinese Academy of Sciences, Shanghai 200031, China. [2]University of Chinese Academy of Sciences, Beijing 100049, China. ✉e-mail: tyang@ion.ac.cn

However, in the new task, each stimulus could be either red or green, and its shape provided information only on the reward probability of the target that had the same color. Therefore, the evidence associated with each shape had to be determined from both the shape and the color of the stimulus before it was accumulated. We observed that the LIP neuronal activity could be decomposed into an early and a late component. The early component encoded the shape weights and additional computation signals, which were independent of the eye movement decisions and represented the sensory-evidence transformation. The late component, in the familiar form of ramping activity, reflected the accumulation of evidence and the formation of decisions. By showing that both the sensory-to-evidence transformation process and the evidence accumulation process are represented in LIP, we conclude that the evidence accumulation signal in LIP arises locally.

## Results

### Behavior

We trained two monkeys with a probabilistic reasoning task (Fig. 1a). The monkeys chose between a red target and a green target based on six sequentially presented stimuli. Either the red or the green target would appear in the recorded neuron's receptive field in a trial. Each stimulus was presented for 333 ms with a 133 ms interval in between. The stimulus set included six shapes, and each could be either red or green. The stimuli in each trial were independently and randomly sampled with replacement from the stimulus set. Each shape was assigned a unique weight. The targets may yield a big or a small amount of juice reward. The probability of getting a large juice amount by choosing a target was determined from the summed weights of the shapes with the same color in the stimulus sequence:

$$P\left(\mathrm{Red}|S_{r_1},\dots S_{r_n}\right) = 10^{\sum_{i=1}^{n}W_{\mathrm{red}_i}} / \left(1 + 10^{\sum_{i=1}^{n}W_{\mathrm{red}_i}}\right), \quad (1)$$

and

$$P\left(\mathrm{Green}|S_{g_1},\dots S_{g_m}\right) = 10^{\sum_{i=1}^{m}W_{\mathrm{green}_i}} / \left(1 + 10^{\sum_{i=1}^{m}W_{\mathrm{green}_i}}\right), \quad (2)$$

where $n$ and $m$ are the numbers of red and green shapes displayed in a trial ($n + m = 6$), $P(\mathrm{Red}|S_{r_1},\dots S_{rn})$ is the reward probability of the red target given all the red shapes $S_{r_1},\dots S_{rn}$, whose weights are $W_{\mathrm{red}_i}$, and $P(\mathrm{Green}|S_{g_1},\dots S_{gm})$ is the reward probability of the green target given all the green shapes $S_{g_1},\dots S_{gm}$, whose weights are $W_{\mathrm{green}_i}$. Positive weights indicated larger than 0.5 probabilities of getting a big amount of juice.

The monkeys had to maintain fixation when viewing the stimuli and choose the better target by saccading toward it when the fixation point was turned off. The reward was delivered based on the reward probability of the chosen target for Monkey M. For monkey H, we delivered a juice reward only when the target with a higher reward probability was chosen to encourage a better performance. Regardless of the different reward schemes in the two monkeys, decision making in this task required both monkeys to evaluate evidence associated with each shape based on its shape and color and accumulate evidence accordingly.

The task required the monkeys to accumulate evidence based on both the shape and the color of the stimuli. A shape in green with a positive weight supports the green choice, but a green shape with a negative weight supports the red choice. Likely, a shape in red may support either the green or the red target depending on the sign of its assigned weight. The choice should depend on the difference between the total weight of the red shapes ($\sum W_{\mathrm{red}}$) and that of the green shapes ($\sum W_{\mathrm{green}}$). This choice pattern was found in both monkeys: they were more likely to choose the red target when $\sum W_{\mathrm{red}}$ was larger than $\sum W_{\mathrm{green}}$ (Fig. 1b). Furthermore, the monkeys did not only use the red

or the green shapes to perform the task, as their choices were correlated with the total weights of shapes in both colors (Supplementary Fig. 1).

The monkeys learned the weights assigned to each shape, although imperfectly. With logistic regression, we estimated the leverage of each shape on the monkeys' choices. The regression coefficients, termed subjective weights[16], mostly correlated with the assigned weights, especially for monkey M (Fig. 1c, top panel). Monkey H did not learn the shapes with negative weights well (Fig. 1c, bottom panel). Those were the shapes that indicated the corresponding target would have a small reward probability. Nevertheless, 5 out of the 6 shapes were significantly different from 0, and the subjective weights were overall in the same ranking order as the assigned weights. To account for monkey H's performance, we used the subjective weights in the neuronal analyses instead of the weights that we assigned to the shapes. Using actual weights leads to qualitatively similar results. All analyses were performed and presented for each monkey individually to ensure our conclusions are valid across the two monkeys.

Finally, all stimuli in the sequence were used by the monkeys for their decisions. We looked at the effects of the stimuli by their order on the monkeys' choices with another logistic regression model (see "Methods"). All stimuli in the sequence exerted significant influence on the monkeys' choices, suggesting that they were integrated to form choices (Fig. 1d).

### LIP activity represented accumulated evidence

After the monkeys learned the task, we recorded single-unit activities from LIP. Only well-isolated neurons with persistent activity and spatial selectivity during the delay period in a memory saccade task were used for the analyses. We collected data from 207 neurons from two monkeys (monkey M: 115, monkey H: 92). We identified the neurons' receptive field location with the memory saccade task. During the recording, one of the targets was located in the receptive field ($T_{\mathrm{in}}$) of the neuron being recorded, and the other target was in the opposite visual hemifield ($T_{\mathrm{out}}$). The color of $T_{\mathrm{in}}$ and $T_{\mathrm{out}}$ was random trial-by-trial.

Consistent with what has been reported previously[15,16], the LIP neurons' activities exhibited a ramping pattern and encoded the accumulated evidence. We quantified the accumulated evidence with the difference between $T_{\mathrm{in}}$ and $T_{\mathrm{out}}$'s total subjective weights ($\sum SW_{\mathrm{in}} - \sum SW_{\mathrm{out}}$). We grouped all trials by choice, divided the trials of each choice into quintiles according to the accumulated evidence in each epoch, and plotted the average responses of each quintile (Fig. 2a, c). The responses in trials with $T_{\mathrm{in}}$ choices were overall larger. Within each choice group, the trials with more evidence supporting $T_{\mathrm{in}}$ elicited higher responses. We further calculated the average firing rate between 300 and 800 ms after the stimulus onset in each epoch and plotted it as a function of the accumulated evidence separately for the $T_{\mathrm{in}}$ and $T_{\mathrm{out}}$ choices. While the slopes of the fitting lines for the $T_{\mathrm{out}}$ trials were stable across the epochs, those for the $T_{\mathrm{in}}$ trials tended to decrease, to the point that they became statistically zero in the last epoch (Fig. 2b, d and Supplementary Fig. 2).

### LIP encoded weights

So far, the current results mostly replicate the previous findings of LIP neuronal responses reflecting the accumulation of evidence[16]. The main purpose of the study is, however, to find out whether LIP is involved in computing evidence from sensory inputs. In this task, the evidence provided by each stimulus is determined from both the stimulus shape and the stimulus color: the shape is associated with a weight that affects the target's reward probability, and the color indicates the relevant target. Both shape weight and color are orthogonal to eye movement decisions. Any representations of shape weight and color should precede that of evidence if they are used for computing evidence.

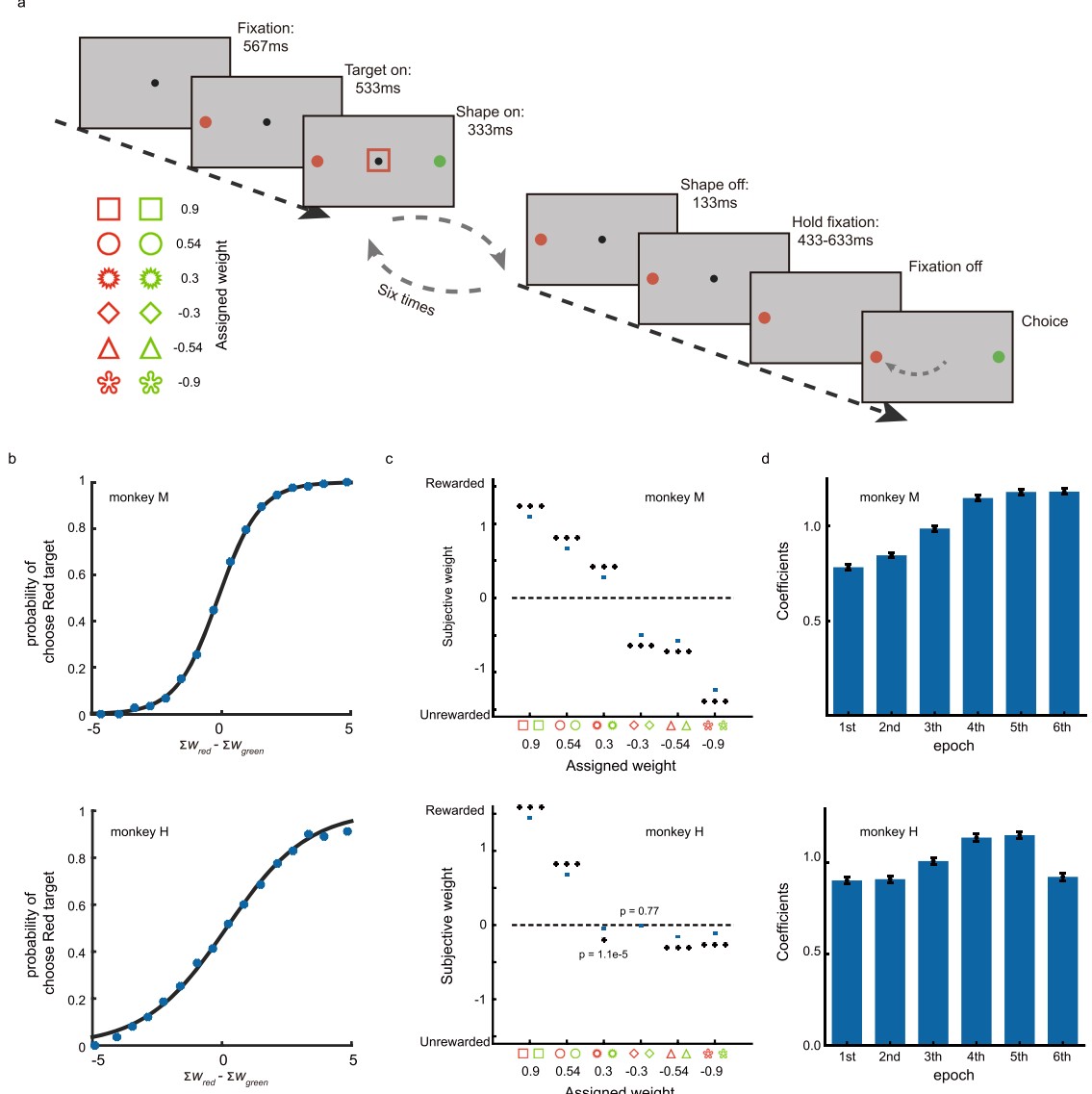

**Fig. 1 | The task and the monkeys' performance. a** The schematic diagram of the probabilistic reasoning task. Targets are displayed in the periphery at 567 ms after the fixation acquisition. After another 533 ms, six stimuli are presented sequentially at the center. The stimuli are randomly selected from a pool of six shapes; each assigned with a weight (bottom left). The shapes can be either red or green, and the reward probability of each target is determined by the total weights of the shapes with the same color. **b** Psychometric curve. The monkeys tended to choose the red target when the total evidence supported the red target. **c** The subjective weight for each shape. The shapes with a positive weight favored the choice toward the target with the same color. The error bars indicating S.E. are too small to be seen for most data points. * $p < 0.001$, *** $p < 1e-10$, two-tailed t test. **d** Effects of stimulus order. Data are presented as mean values +/− S.E. Plots in the upper and the lower panel in (**b**–**d**) include 56,074 and 42,516 trials from monkeys M and H, respectively.

We first looked at the encoding of shape weight. We plotted the population PSTH to each shape averaged across trials and epochs (Fig. 3a). In both monkeys, LIP neurons' responses were suppressed by the shapes with a positive weight, but only transiently. In approximately 1000 ms after the shape onset in both monkeys, all traces merged, and the representation of shape weight disappeared (Fig. 3a). Each shape lasted 333 ms on the screen, but it affected LIP neurons responses well beyond its presentation period until after the next shape. We disentangled the effect of each shape's weight on LIP responses with a Poisson generalized linear model (GLM) (see "Methods"). The kernel for shape weight from the GLM captures the dynamic relationship between shape weight and the neurons' spiking probability. We plotted the kernel for each neuron in Fig. 3b and the population average kernel in Fig. 3c. Again, the GLM revealed that most neurons' responses were negatively correlated with shape weight after the stimulus onset, denoted by the blue color in Fig. 3b and the early

dip below 0 in the population kernels in Fig. 3c (negative: 76/115 in monkey M, 53/92 in monkey H; positive: 1/115 in monkey M, 10/92 in monkey H). The kernel weights faded away gradually to 0, suggesting that shape weight only had a transient effect on the neurons' responses.

To compare LIP neurons' weight encodings against their evidence encoding, we extracted the neurons' response kernels for evidence with the same Poisson GLM. Plotted in Fig. 4a are the neuron's evidence kernels, and the neurons are indexed in the same order as in Fig. 3b. The population average kernel is shown in Fig. 4b. The majority of LIP neurons had positive evidence kernel weights, denoted by the warm colors, meaning that the neurons had larger responses when the evidence was in favor of $T_{in}$. Moreover, unlike the encoding of shape weight, the encoding of evidence did not fade away and was persistent. These results are consistent with Fig. 2, both indicating a representation of accumulated evidence in LIP.

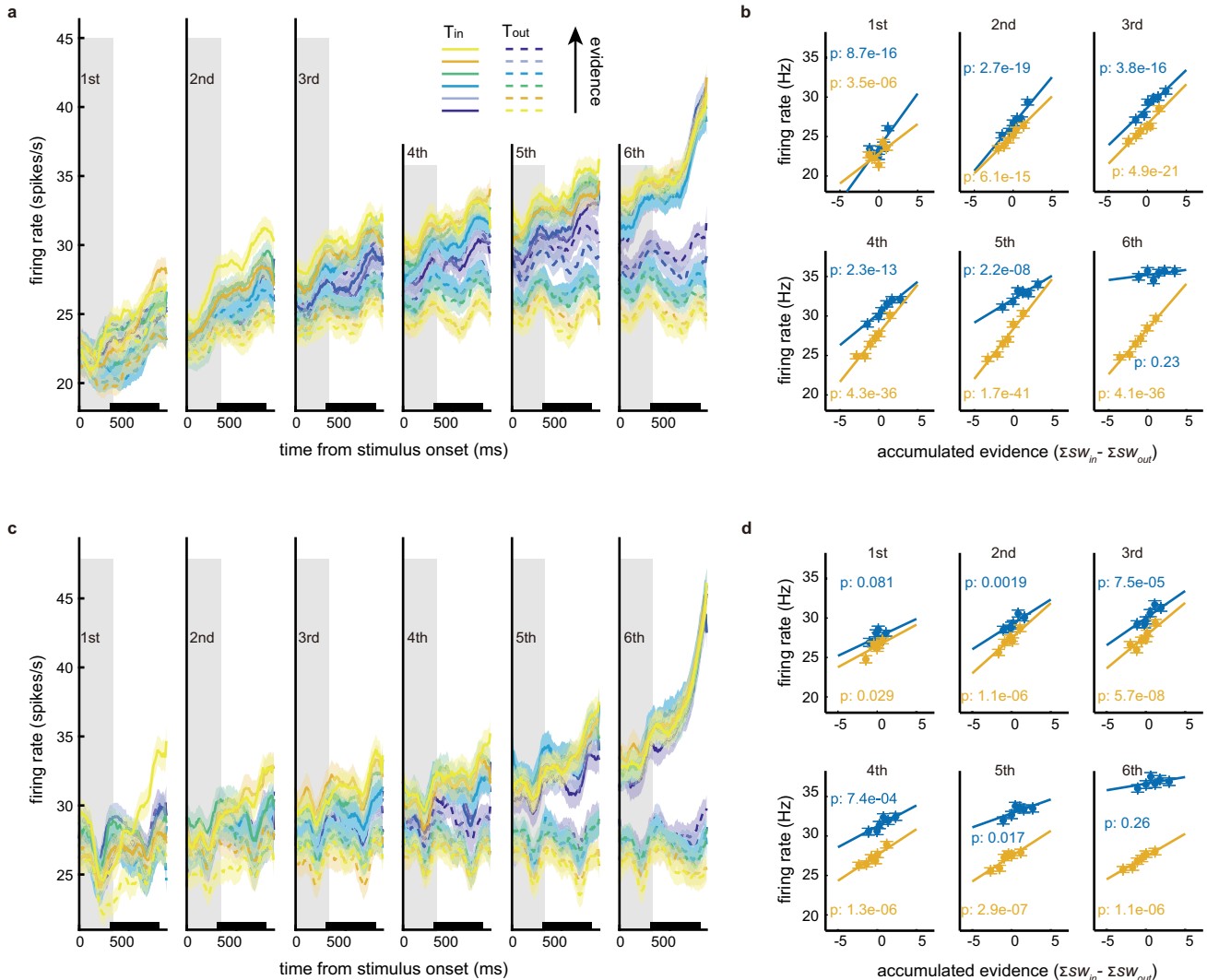

**Fig. 2 | LIP responses represented accumulated evidence. a** The population-averaged peri-stimulus time histograms (PSTHs) of the 115 neurons from monkey M aligned to the stimulus onset in each epoch. The gray areas indicate the period when the stimulus is on the screen. The traces correspond to the quantiles sorted by the accumulated evidence ($\sum SW_{in} - \sum SW_{out}$) in each epoch, indicated by different colors. Solid and dashed lines indicate the trials with $T_{in}$ and $T_{out}$ choices, respectively. Curve thickness depicts S.E. across units. **b** The firing rates (averaged during the period indicated by the black bar on the x-axis in **a**) of the 115 neurons from monkey M are plotted against the accumulated evidence in each epoch. Blue and orange are the trials with $T_{in}$ and $T_{out}$ choices, respectively. The lines are the fitting lines with the p values indicated (two-tailed tests without multiple comparisons adjustments). Trials are binned into six groups according to accumulated evidence. **c** Same as (**a**), but for the 92 neurons from monkey H. **d** Same as (**b**), but for the 92 neurons from monkey H.

In terms of the timing, we found that the representation of shape weight preceded that of evidence. Based on the GLM, we estimated the onset latency of both shape weight and evidence encoding for each neuron (see Methods). The average latency of the shape weight encoding across neurons was significantly shorter than that of the evidence encoding (Fig. 4c, Monkey M: 211.95 ± 8.78 ms vs. 681.72 ± 34.19 ms, Monkey H: 237.66 ± 13.42 ms vs. 940.31 ± 39.01 ms), which is consistent with the idea that the LIP activity reflects the transformation from sensory inputs to the evidence for eye movements. Although the exact values of latency may differ significantly with different ways to determine the latency, and our particular method may overestimate the latencies, the latency order between the two encodings remains robust.

Despite the timing difference, the encoding of shape weight and evidence was strongly coupled in individual neurons. This is apparent in Fig. 4a, where the evidence kernels appear well ordered even though the neurons are indexed with their shape weight kernels. We further used the GLMs to calculate the coefficients of partial determination

(CPD) of shape weight and evidence to estimate the fraction of the response variance uniquely explained by each variable (see "Methods"). A large CPD indicates a strong contribution to explaining the response variance that other variables cannot explain. There was a clear positive correlation between the two CPDs (Fig. 4d): the neurons that encoded evidence strongly also tended to encode shape weight. A single pool of LIP neurons encoded both variables.

## The stimulus-to-evidence transformation

The shape weight alone is not sufficient for LIP neurons to compute a shape's evidence for the eye movement decision. The neurons also need to know the shape's color and $T_{in}$'s color. With these three pieces of information, there are two routes to compute evidence. One is to first determine whether the shape's color is the same as $T_{in}$'s color (color consistency). The evidence is the shape weight if their colors match, and the minus shape weight if they do not (Fig. 5a, route 1). The other route is to first combine the shape's weight and color to compute the evidence for the target with the same color (color evidence), and

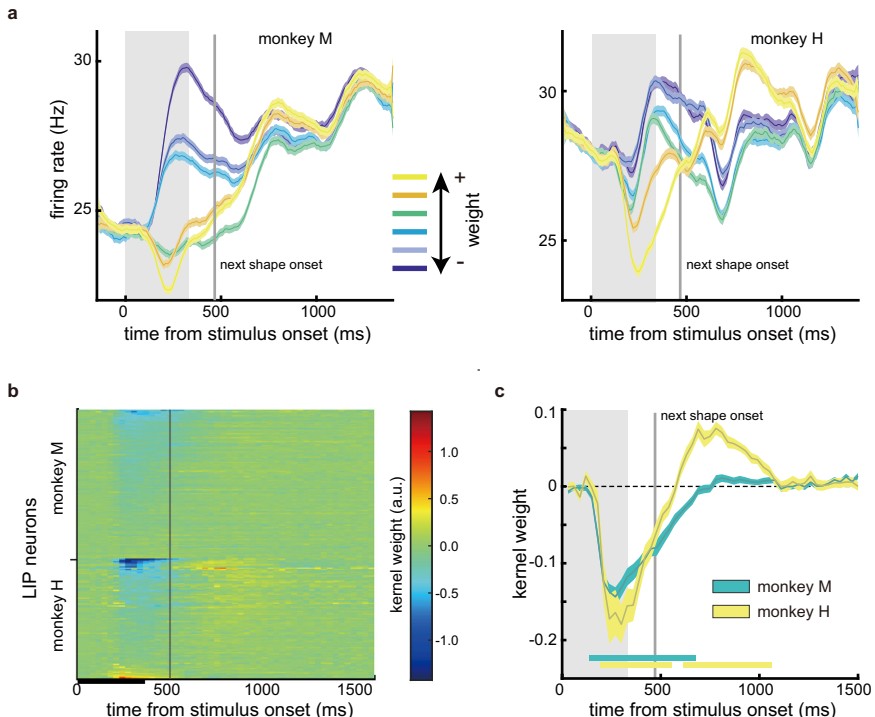

**Fig. 3 | LIP neurons encoded shape weights. a** The PSTH (left: monkey M, right: monkey H) aligned to the onset of the shapes, averaged across the epochs and the trials. The traces are sorted by shape weight. **b** Response kernels for shape weight based on the Poisson GLM. Each row is a neuron, and the colors denote the gain. Red and blue indicate positive and negative kernel weights, respectively. Most neurons show negative kernel weights (blue, monkey M: 76 out of 115, monkey H: 53 out of 92), indicating that the shapes with a positive weight suppress LIP responses.

Neurons are sorted by average kernel weights across time, separately for each monkey. The black bar on the time axis indicates the period when the stimulus is on the screen. **c** Population kernels for shape weight. The bars along the abscissa indicate the periods when the kernel weights are significantly different from 0 (two-tailed t tests, $p < 0.01$). The curve width in **a** and **c** denotes S.E. across units. Shaded areas indicate the period when the stimulus is on the screen. The grey vertical lines indicate the onset of the next stimulus.

then determine the evidence based on whether this target is $T_{in}$ or $T_{out}$ (Fig. 5a, route 2).

To examine whether any of the two computation routes were reflected by the LIP activity, we investigated the representation of the intermediate variables, i.e., color evidence and color consistency, in LIP, again with Poisson GLM (see Methods). The results were different between the two monkeys. For monkey M, color consistency, but not color evidence, was represented in its LIP (Fig. 5b, d). 47 out of 115 neurons showed significant selectivity for color consistency. Their responses were lower when the colors of the stimulus and $T_{in}$ matched. Just as shape weight, color consistency only had a transient influence on the LIP responses. The average latency of color consistency encoding was between those of shape weight and evidence (Fig. 5f, weight: $211.95 \pm 8.78$ ms, consistency: $357.50 \pm 14.89$ ms, evidence: $681.72 \pm 34.19$ ms). In addition, at the single neuron level, the encoding of color consistency is coupled with the encoding of both shape weight (Fig. 5g) and evidence (Fig. 5h), suggesting the three variables were represented by the same neurons. The encoding of color evidence was not found in monkey M (Fig. 5c, e). The results suggested that LIP in monkey M carried out the computation according to route 1.

Monkey H, however, showed a distinct pattern. Color consistency was encoded in monkey H's LIP, but only by a small number of neurons (Fig. 5b, 2 out of 92 neurons) and only weakly at the population level (Fig. 5d). On the other hand, color evidence was encoded robustly at the population level (Fig. 5e). At the individual neuron level, although many neurons exhibited a trend of encoding color evidence (Fig. 5c), only one out of 92 neuron's encoding was significant, preventing us from assessing the encoding latency and whether the encoding of color evidence correlated with the encoding of the other variables. Overall, the data favors the hypothesis that LIP in monkey H used route 2 to compute evidence.

Although the LIP activity patterns in the two monkeys suggested that they computed evidence in two different routes, via color consistency in monkey M and via color evidence in monkey H, the results from both monkeys consistently showed that the early component of LIP responses reflected the intermediate variables in the transformation from the sensory inputs to the evidence for eye movement decisions. These variables are orthogonal to eye movements but encoded by the same neurons that accumulate evidence for eye movement decisions.

**Accumulation of difference, not the difference of accumulation**
One final scenario that has not been discussed so far is that the LIP neurons might only accumulate a shape's evidence when the shape's color matches its response field target's, and the final decision might be based on the comparison between two populations of neurons, each accumulates evidence for $T_{in}$'s and $T_{out}$'s color, respectively. Therefore, we investigated whether LIP activity reflects a decision process based on the competition between the evidence accumulated separately for the two targets or on the accumulation of each shape's evidence favoring one target versus the other.

We first compared how the evidence associated with each stimulus affected the LIP neurons' responses when its color matched $T_{in}$'s ($SW_{in}$) and when it matched $T_{out}$'s ($SW_{out}$), again with the Poisson GLM. If the evidence for each colored target is integrated separately, LIP neurons would only accumulate the evidence when the stimulus color matched $T_{in}$'s, and $SW_{out}$ should have negligible impact on their responses. This is not what we observed. Figure 6a shows the population response kernels for $SW_{in}$ and $SW_{out}$. After the initial dip shared by both, likely caused by the negative shape weight encoding, the neurons' responses became positively correlated with $SW_{in}$ and

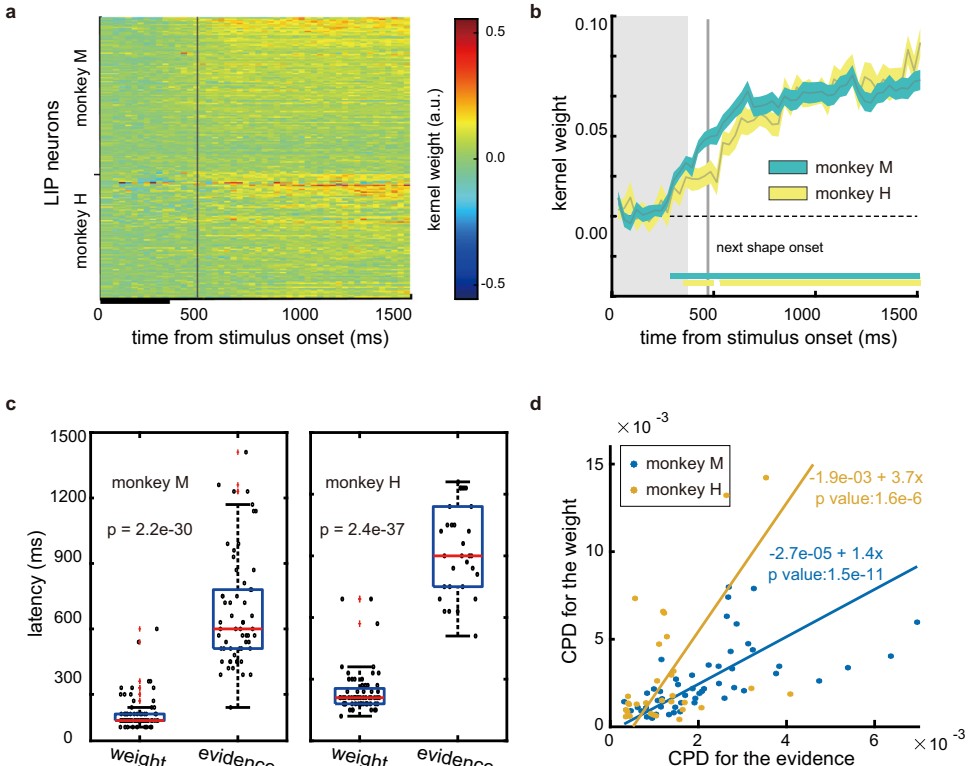

**Fig. 4 | LIP neurons encoded evidence. a** Response kernels for evidence based on the Poisson GLM. Each row is a neuron. Red and blue indicate positive and negative kernel weights, respectively. Many neurons show positive kernel weights (monkey M: 58 out of 115, monkey H: 32 out of 92), indicating that evidence in favor of $T_{in}$ increased their responses. Neurons are indexed in the same order as in Fig. 3b. **b** Population evidence kernels. The curve width denotes S.E. across units. The bars along the abscissa indicate the periods when the kernel weights are significantly different from 0 (two-tailed t tests, $p < 0.01$). The grey vertical lines in **a** and **b** indicate the onset of the next stimulus. **c** The encoding latencies of shape weight and evidence. Each dot denotes a neuron. Only neurons with significant encodings are included (monkey M, weight: 77, evidence: 58; monkey H, weight: 63, evidence: 32). The bottom and top edges of the box are the 25th and 75th percentiles, respectively, the central red line indicates the median, and '+' denotes outliers. The $p$ values are from two-tailed independent t tests, indicating a significant difference between the encoding latencies of shape weight and evidence. **d** The CPDs for shape weight and evidence. Each dot denotes a neuron and the color denotes the monkey. Only neurons with significant encodings of both variables are included (monkey M: 52, monkey H: 31). The lines are the linear models fitted to the data points with a model II regression.

negatively correlated with $SW_{out}$. The kernel weights were roughly symmetric for both $SW_{in}$ and $SW_{out}$ in the later period, indicating that the LIP neurons' responses were equally, although in opposite directions, affected by a shape's evidence, no matter whether the shape had the same color as $T_{in}$ or not.

We further studied the encodings of $SW_{in}$ and $SW_{out}$ at the level of single neurons. Based on the neurons' encoding latency for evidence (Fig. 4c), we chose a time window at 750–1500 ms (monkey M) and at 1000–1750 ms (monkey H) after the stimulus onset for this analysis. We computed each neuron's average kernel weights within these time windows. A large fraction of the LIP neurons encoded both $SW_{in}$ and $SW_{out}$ (blue and black markers in Fig. 6b, 83/115 in monkey M, 54/92 in monkey H), with many of them are statistically indistinguishable (black markers in Fig. 6b, 41/115 in monkey M, 35/92 in monkey H). Only a small number of neurons encoded just $SW_{in}$ (red markers in Fig. 6b, 13/115 in monkey M, 18/92 in monkey H) or $SW_{out}$ (green markers in Fig. 6b, 10/115 in monkey M, 9/92 in monkey H). Overall, the encodings of $SW_{in}$ and $SW_{out}$ were similar but in opposite signs. The results further confirmed that the LIP neurons accumulated evidence difference between $SW_{in}$ and $SW_{out}$ regardless of whether the shapes' color matched with the color of $T_{in}$ or not. The evidence was not accumulated separately for each color.

## Discussion

Using a probabilistic reasoning task, we show that the ramping activity observed in LIP during decision making can be segmented into an early

component and a late component. While the later and sustained component reflected the accumulated evidence for eye movement decisions, the early but transient component represented the variables in the transformation from sensory inputs into the evidence before accumulation. The parsimonious interpretation of these results is that LIP is where evidence is evaluated and accumulated towards a decision.

Although the early components of LIP responses reflected the sensory inputs, they were unlike shape selectivity[18], color selectivity[19], or motion selectivity[20–22] previously reported for the LIP neurons. In those studies, the selectivities were measured by stimuli in the response field of the neurons, and it is not clear how these selectivities are related to evidence accumulation or decision making. In the current study, we focused on the neurons that showed delay activity in a delay saccade task, and in all aspects of their response properties, they were the same type of neurons that were reported in earlier decision making studies[2,15,16]. The stimuli were always presented at the center, while the response fields of the neurons were peripheral.

Our findings suggest that LIP computes and accumulates evidence based on arbitrary sensory inputs. In several previous studies of LIP with random dots motion stimuli, it was found that the dots' motion information was encoded in LIP before a motor plan was formed[23,24]. However, these results are not sufficient for arguing for LIP's general role in evidence accumulation. This is because LIP sits in the dorsal stream of visual processing for motion and carries motion information[21,25]. The observed motion information may be

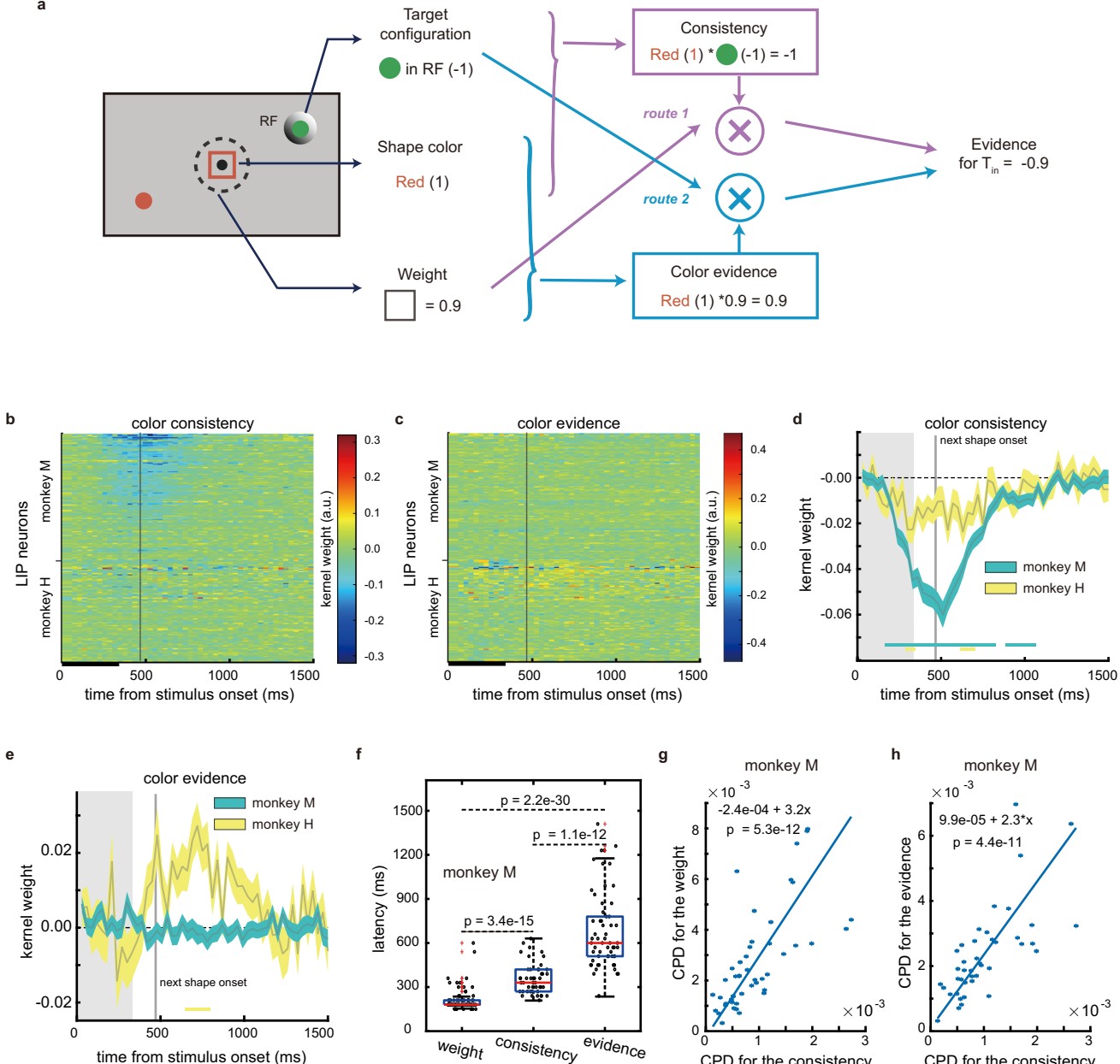

**Fig. 5 | LIP activity reflected stimulus-to-evidence transformation. a** Two routes to compute a stimulus's associated evidence for $T_{in}$. In route 1 (purple lines), the color consistency is calculated and further combined with the shape's weight to form evidence. In route 2 (blue lines), the evidence supporting the target that matches the stimulus's color (color evidence) is first evaluated, and evidence is determined based on whether the target is $T_{in}$ or not. **b–c** The response kernels for color consistency (**b**) and color evidence (**c**). Same format as Fig. 3b. In monkey M, many neurons show significant color consistency kernel weights (47 out of 115), and only 1 neuron shows significant color evidence kernel weights. In monkey H, only 2 and 1 out of 92 neurons show significant kernel weights for color consistency and color evidence, respectively. **d–e** Population averaged kernels for color consistency (**d**) and color evidence (**e**). The bars along the abscissa indicate the periods when

the kernel weights are significant (two-tailed t tests, $p < 0.01$). The curve width denotes S.E. across units. The grey vertical lines indicate the onset of the next stimulus. **f** The encoding latencies of color consistency, weight and evidence in monkey M. Each dot denotes a neuron. Only neurons with significant encodings are plotted (weight: 77, consistency: 48, evidence: 58 out 115). Same format as Fig.4c. *P* values are from two-tailed t tests. **g** The CPDs for weight and consistency correlate at individual neuron level in monkey M. $p = 5.3e{-}12$, two-tailed t test. **h** The CPDs for evidence and color consistency correlate at individual neuron level in monkey M. $p = 4.4e{-}11$, two-tailed t test. Only neurons with significant encodings of both variables are plotted ($n = 47$ (**g**) and 42 (**h**) out 115). The lines are fitted with model II regressions in **g** and **h**. The significance is evaluated with the Ricker procedure.

conveniently used in the motion direction discrimination tasks, but one cannot assume other types of sensory information are also encoded in LIP and used for decision making based on these studies. In contrast, our study demonstrated LIP's representations of the sensory information and extra computations before evidence accumulation. In addition, the strengths of the representations of the early sensory component and the late evidence component, evident from the early negative peak and the late sustained activation in Fig. 3c and 4a, were comparable, suggesting that the encoding of the early component was not weak or secondary. Therefore, our findings argue strongly for the hypothesis that LIP computes and accumulates evidence based on arbitrary sensory inputs, which in our case required multiple steps of computation before their associated evidence can be accumulated.

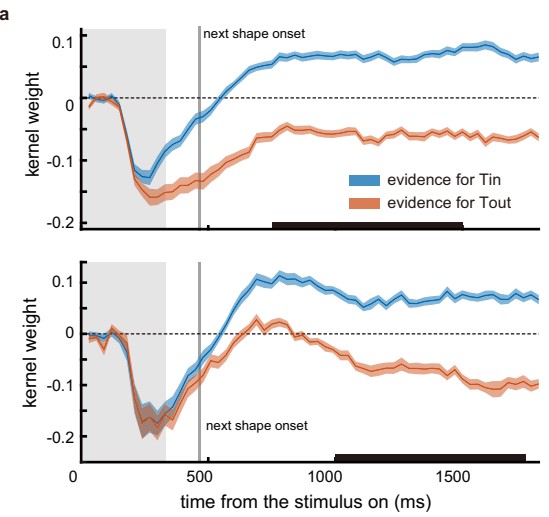

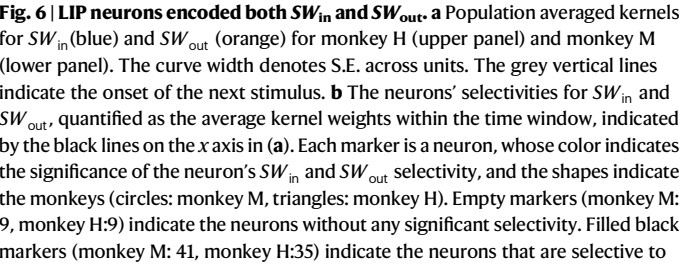

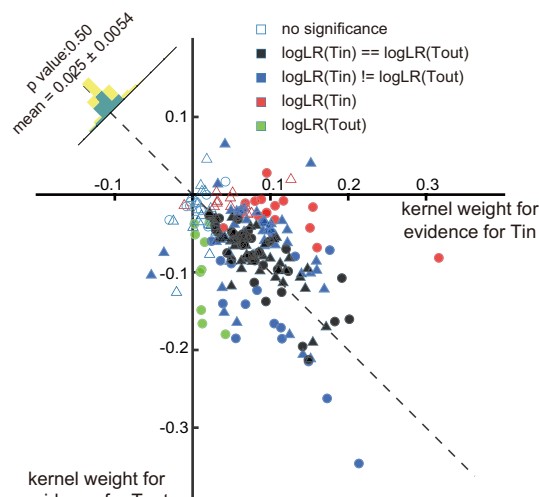

**Fig. 6 | LIP neurons encoded both $SW_{in}$ and $SW_{out}$. a** Population averaged kernels for $SW_{in}$ (blue) and $SW_{out}$ (orange) for monkey H (upper panel) and monkey M (lower panel). The curve width denotes S.E. across units. The grey vertical lines indicate the onset of the next stimulus. **b** The neurons' selectivities for $SW_{in}$ and $SW_{out}$, quantified as the average kernel weights within the time window, indicated by the black lines on the *x* axis in (**a**). Each marker is a neuron, whose color indicates the significance of the neuron's $SW_{in}$ and $SW_{out}$ selectivity, and the shapes indicate the monkeys (circles: monkey M, triangles: monkey H). Empty markers (monkey M: 9, monkey H:9) indicate the neurons without any significant selectivity. Filled black markers (monkey M: 41, monkey H:35) indicate the neurons that are selective to both, and the selectivities for $SW_{in}$ and $SW_{out}$ are not significantly different other than the sign. Blue markers (monkey M: 42, monkey H: 19) indicate the neurons that are selective to both, but the selectivity is significantly different. Red markers (monkey M: 13, monkey H:18) indicate the neurons only selective to $SW_{in}$. Green markers (monkey M: 10, monkey H: 9) indicate the neurons only selective to $SW_{out}$. The dashed line is the anti-diagonal, and the histogram shows the distribution of the difference between the neurons' selectivities for $SW_{in}$ and $SW_{out}$ in monkey M (blue) and H (yellow), whose means are not significantly different from 0 (monkey M: $p = 0.65$, monkey H: $p = 0.83$, two-tailed t test). The mean and the $p$ value for the combined monkeys' data are indicated in the histogram.

The representation of both the evidence accumulation and the computation before accumulation has not been demonstrated in any other brain areas. In particular, our results provide an interesting contrast against the findings from the prefrontal cortex. In a recent study, Lin et al. used a similar but simpler task to study evidence accumulation in the prefrontal cortex[5]. Although the dorsolateral prefrontal cortex (DLPFC) was also found to accumulate evidence for eye movements, it only weakly encoded stimuli's color weights, which were quantities comparable to shape weight in the current study, and a correlation between the encoding of color weight and evidence at individual neuron level was absent. Even though DLPFC neurons are well known for their heterogeneity[26], their responses did not fully capture the transformation from sensory inputs to accumulated evidence. In addition, the signal for accumulated evidence in the DLPFC had a later onset than what was found in LIP in the current study. Therefore, the accumulated evidence signal in LIP is unlikely to come from the DLPFC. Between the two areas, LIP is a more likely candidate where evidence accumulation occurs.

Although we interpret the early component of LIP responses as sensory, it is relative to the later accumulated evidence and eye movement signals. Shape weight represented in LIP is already an intermediate variable between raw sensory inputs and the following computations. The shape weight signals may arise from the orbitofrontal cortex (OFC). Shape weight is a quantity tied to value, and the OFC has been shown to encode value associated with visual stimuli[5,27,28].

The current literature cannot explain why the early LIP responses negatively encoded the shapes' weight regardless of their color. Because shape weight is related to targets' reward probability, the results mean that stimuli that led to larger reward probabilities suppressed LIP neurons' responses. This is opposite to the studies that showed that the LIP activities were positively correlated with choice reward[29,30], suggesting that the weight encoding cannot be explained by choice reward. The negative encoding also cannot be explained by salience[31], which may be quantified as the absolute value of shape

weight in our experiment. The weight encoding in LIP was signed: positive weights induced suppression of LIP responses, and negative weights increased the neurons' responses. We suspect that the negative encoding reflected the nature of the inputs from the brain area where LIP receives this information. Depending on where information comes from and how information is encoded in the source brain area, we may observe different encoding patterns of sensory inputs in LIP. The encoding may be consistent with, orthogonal to, or even opposite to the eye movement signals in LIP. In a recent study, LIP neurons were found to encode evidence negatively in a face discrimination task, providing another example where the encoding of sensory inputs was seemingly contradictory to evidence accumulation[17].

The early negative weight encoding may also explain the "dip" in the LIP activity right after the motion onset. The dip has been observed in many previous random-dot studies. It was proposed that the dip may be caused by the lateral inhibition within the LIP network or a general normalization mechanism[32]. This explanation is not supported by our data, because the weight encoding that we observed had a short latency and preceded the encoding of evidence. In addition, the dip occurred after each stimulus in the trial sequence (Fig. 1a, b and Supplementary Fig. 3), which excludes the possibility that the dip was associated with any particular events related to the starting of the accumulation. The dip also could not be explained by a temporal attraction of attention away from the eye movement targets and toward the stimuli at the fixation[33]. If this was the case, we would expect the dip reflected stimulus salience, which should be high for shapes with both large positive weights and large negative weights. The negative encoding of the weights contradicts this scenario.

In summary, we demonstrated that LIP activity represented the computation of the sensory-stimulus transition in addition to the evidence accumulation during decision making. The representation of both stages of computation suggests that the evidence accumulation signal in LIP may arise locally, and thus provides a strong argument for LIP's important role in evidence accumulation during decision making.

## Methods

### Subjects and materials

Two adult male rhesus macaques (Macaca mulatta) were used in the study (M and H). They weighted on average 10–11 kg during the experiments. All experimental procedures were approved by the Animal Care Committee of Shanghai Institutes for Biological Sciences, Chinese Academy of Sciences (Shanghai, China).

During training and recording sessions, the monkeys were seated in a primate chair. Visual stimuli were presented on a 23.6-inch video monitor, which was placed at a 60 cm distance from the monkey. The behavioral paradigm was controlled through custom code with the MatLab-based MonkeyLogic Version 1. The eye movements were tracked with a high-speed infrared camera at a sampling rate of 500 Hz (Eyelink 1000, SR-Research). Juice or water reward was provided as rewards based on the monkeys' preference. The liquid delivery was controlled by a computer-controlled solenoid. The monkeys drank ~200–350 ml per experimental session.

### Probabilistic reasoning task

Monkeys started each trial by fixating on a central fixation point (FP) (0.3° in diameter). After 567 ms, a red and a green target (0.2° in diameter) were displayed on the left and right sides of the FP, equidistant from the FP. During the recording sessions, one of the targets was in the response field of the neuron being recorded, and its color varied trial by trial. After 533 ms, six visual stimuli were shown at the center of the screen sequentially. Each was displayed for 333 ms and followed by a 133 ms delay. After the last shape disappeared, there was a variable delay (randomly selected from 433/533/633 ms). Afterward, the FP was turned off, instructing the monkey to report its choice by making a saccade to one of two targets and fixating on the target for 283 ms. The reward was delivered based on the chosen target's reward probability.

The stimuli were randomly sampled with replacement from a pool of six shapes. Each shape was assigned a unique weight. The stimulus color was randomly chosen between red and green. The color indicated the target that the stimulus carried information about. Each target yields either a large reward or a small reward, and the probability was determined by the total weight of the shapes with the same color that appeared in the trial (Eqs. 1 and 2). For monkey M, a large reward (120 µl/drop, 2 drops) was delivered with a probability of $P(\text{target}_{\text{chosen}})$, and a small reward (60 µl/drop, 1 drop) was delivered with a probability of $1 − P(\text{target}_{\text{chosen}})$. We rewarded monkey H only when the target with a larger reward probability was chosen. A large reward (200 µl/drop, 2 drops) was delivered with a probability of $P(\text{target}_{\text{chosen}})$, and a small reward (100 µl/drop, 1 drop) was delivered with a probability of $1 − P(\text{target}_{\text{chosen}})$.

### Surgery procedures and recordings

The monkeys received a chronic implant of a titanium headpost with standard procedures before the training. After their performance reached a satisfactory level, we performed a second surgery to implant an acrylic recording chamber over the intraparietal sulcus, which allowed electrophysiological recording from LIP. The inner diameter of the cylindrical chamber was 19 mm. All surgery procedures were done under aseptic conditions. The monkeys were sedated with ketamine hydrochloride (5–15 mg kg$^{-1}$, i.m.) and anesthetized with isoflurane gas (1.5–2%, to effect). Their body temperature, heart rate, blood pressure, and $CO_2$ were monitored during the surgeries.

### Electrophysiology

A plastic grid (1 mm spacing) was placed inside the chamber to precisely locate the targeted brain areas. Single tungsten electrodes (Alpha Omega) were used for recordings, and electrodes were accurately placed with a microdrive (Electrode Positioning System, Alpha Omega). The neural responses were collected with an AlphaLab SnR System 2.0.4.5 (Alpha Omega).

In each session, one or two single electrodes were placed in the ventral division of LIP (LIPv), which was located based on structural magnetic resonance imaging and transitions of white and gray matter during penetrations. Action potential waveforms were isolated online using a window discriminator or sorted offline (Offline Sorter Application Version 4.5.0). Only units with well-isolated waveforms were recorded and used for analysis.

Units with persistent activity and spatial selectivity in a memory-guide saccade task were used for further analysis. In the memory-guide saccade task, a target appeared shortly (200 ms) in the periphery while the monkey fixated the central fixation point. After a variable delay period (uniformly selected from 533/633/733 ms), the fixation point disappeared, instructing the monkey to make a saccade toward the target. The receptive field was defined as the target position where the unit showed the maximal responses during the delay period. During the probabilistic reasoning task, one of two targets was placed in the receptive field, and the other was in the opposite visual hemifield. We collected data from 207 neurons (monkey M, 115 neurons, monkey H 92 neurons).

### Behavioral analyses

Logistic regression was applied to determine how each shape affected the choice. The probability of choosing the red target is a function of the sum of leverages, $Q$, provided by the presented stimuli:

$$P(\text{choice} = \text{red}) = \frac{1}{1 + 10^{-Q}}, \tag{3}$$

$$Q = \beta_0 + \sum_{i=1}^{6} \beta_i N_{\text{red}_i} - \sum_{i=1}^{6} \beta_i N_{\text{green}_i}, \tag{4}$$

where $N_{\text{red}_i}$ and $N_{\text{green}_i}$ indicate how many times shape $i$ appeared in a trial in red and in green, respectively ($i = 1...6$). $\beta_0$ is the bias term. $\beta_{1-6}$ provide the estimates of the weight that the monkey assigned to the shapes and are defined as the shapes' subjective weights (Yang and Shadlen, 2007).

We performed logistic regression to assess whether the stimuli in all epochs affected monkeys' choice as follows,

$$P(\text{choice} = \text{red}) = \frac{1}{1 + 10^{-Q}}, \tag{5}$$

$$Q = \beta_0 + \sum_{i=1}^{6} \beta_n SW_{\text{red}_i} - \sum_{i=1}^{6} \beta_n SW_{\text{green}_i}, \tag{6}$$

where $SW_{\text{red}_i}$ and $SW_{\text{green}_i}$ are the subjective weight of the red or the green shape appearing in epoch $i$. $SW_{\text{red}_i} = 0$ if the stimulus in epoch $i$ is not red, and $SW_{\text{green}_i} = 0$ if the stimulus is not green. $\beta_0$ is the bias term, $\beta_{1-6}$ are the fitting coefficients of the shapes in each epoch (six epochs in total).

### Neural data analyses

We performed all electrophysiological analyses separately for two monkeys.

**The generalized linear model (GLM).** We followed the previously published model that described the probability of spike trains, $p(\mathbf{r}|\mathbf{x}, \boldsymbol{\theta})$, with external variables[34]. The model assumes the spikes follow a Poisson distribution with a time-varying spike rate $\lambda_t$, which results from the linear convolution of the time course of the task variables with their corresponding kernel $k_i$:

$$\lambda_t = e^{\sum_i (k_i * f(x_i))(t)} \tag{7}$$

$$p(\mathbf{r}|\mathbf{x},\mathbf{k}) = \prod_{t=0}^{T} p(r_t|\mathbf{x},\mathbf{k}) \propto \prod_{t=0}^{T} \triangle \lambda_t^{r_t} e^{-\triangle \lambda_t} \qquad (8)$$

where $x_i$ is the $i$th variable, $k_i$ is the corresponding kernel, $(k_i * f(x_i))(t)$ indicates the linear convolution between $f(x_i)$ and $k_i$, $\Delta$ is the time bin size (10 ms), and $T$ is the number of time bins in each trial. $f(x_i)$ is a boxcar function for all variables, which lasts 1500 ms for the variables related to the shapes and 3000 ms for target onset and target configuration. The response function $k_i$ was fitted to minimize $-\log(p(\mathbf{r}|\mathbf{x},\mathbf{k}))$ across the whole trial period with Matlab's build-in function *fminunc*. The model's predicted responses generated from the 5th and the 6th stimulus extend beyond the end of a trial, but only the part before the end of the trial was considered during the fitting. We did not use any regularization or penalize high-valued kernel weights for us to determine the significance of the kernel weights more easily. The fitting results were validated with five-fold cross-validation (Supplementary Fig. 4).

To quantify the encoding significance, we shuffled the external variables across trials and performed the GLM on the shuffled data sets. The significance of the kernel weight at each time point was determined with a two-tailed t test ($p < 0.01$). We calculated the longest duration of the time window during which the kernel weights were significantly different from 0. This procedure was repeated 100 times to obtain the 95th percentile of the longest encoding duration. A neuron was considered to encode a certain variable significantly only if its kernel's longest encoding duration was above this 95th percentile. In addition, a neuron's encoding was considered positive if the average kernel weight during the first significant encoding duration was positive. Negative encoding neurons were similarly defined. The neuron's encoding latency was defined as the beginning of the first significant encoding duration.

In the GLM in Figs. 3–5, the variables that we included were stimulus onset, subjective weight associated with stimulus shape, stimulus color (1: red, −1: green), evidence, color consistency (1: stimulus color = $T_{in}$ color, −1: stimulus color ≠ $T_{in}$ color), color evidence, target onset, and target configuration (1: $T_{in}$ is red, −1: $T_{in}$ is green).

In Fig. 6, stimulus onset, stimulus color, $SW_{in}$, and $SW_{out}$ were used in the GLM. The other stimulus and target variables included in the GLM above were not considered here, because $SW_{in}$ and $SW_{out}$ were calculated with those variables.

For fair comparisons, all comparisons of average latency, kernel weight, and CPD were based on results from the same GLM.

**Coefficient of partial determination (CPD).** The CPD analyses in Fig. 4d and Fig. 5g, h were based on the same GLMs described above. The CPD of variable $i$ and neuron $j$ was calculated as follows:

$$CPD_{i,j} = \frac{SSE(X_{-i,j}) - SSE(X_{all,j})}{SSE(X_{-i,j})}, \qquad (9)$$

where $SSE(X_{all,j})$ and $SSE(X_{-i,j})$ are the sum of the squared estimate of errors of the full model and the model with variable $i$ *shuffled*. This procedure was repeated 100 times, and the CPD plotted in Fig. 4d and Fig. 5g, h were the averages.

**Reporting summary**
Further information on research design is available in the Nature Research Reporting Summary linked to this article.

## Data availability
The data generated in this study have been deposited in the Zenodo database and are accessible at https://doi.org/10.5281/zenodo.6555461.

## Code availability
All custom code used for reported analyses in this study are available at https://github.com/zwzhangi36/sensory-evidence_transformation_in_LIP.

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

## Acknowledgements

We thank Michael Shadlen and Roozbeh Kiani for their comments during the preparation of the manuscript. We thank Zhongqiao Lin, Yang Xie, Chechang Nie, Wei Kong, Lu Yu for their help in all phases of the study. This work was supported by the National Science and Technology Innovation 2030 Major Program (Grant No. 2021ZD0203701), National Key R&D Program of China (Grant No. 2019YFA0709504), and the Strategic Priority Research Program of Chinese Academy of Sciences (Grant No. XDB32070100).

## Author contributions

T.Y. designed the study. Z.Z. and C.Y. collected the data. Z.Z. analyzed the data. Z.Z. and T.Y. wrote the paper.

## Competing interests

The authors declare no competing interests.
