## [Peer Review File · Nature Communications]

Evidence accumulation occurs locally in the parietal cortexEditorial Note: This manuscript has been previously reviewed at another journal that is not operating a transparent peer review scheme. This document only contains reviewer comments and rebuttal letters for versions considered at *Nature Communications*. Mentions of the other journal have been redacted.

REVIEWERS' COMMENTS

Reviewer #1 (Remarks to the Author):

I previously reviewed this manuscript for [another journal]. The authors have addressed all of my concerns from that previous review. However, there is one point I would like to address. In the Introduction (lines 55-57), the authors state "We reason that if LIP inherits evidence accumulation signal from somewhere else, we would unlikely observe computation stages that happen before the accumulation." One could argue that we may still observe such stages, but the underlying "accumulation" signal would just not be used for evidence accumulation and possibly for some other computation in the decision or motor process. It would greatly benefit the reader if the authors addressed this directly.

Reviewer #2 (Remarks to the Author):

Zhang and colleagues present a novel and exciting finding suggesting that the lateral intraparietal area locally accumulates information and computes evidence for decision making. This is a very important finding, especially in light of recent debates about distributed representations in frontal cortex that allow for little systematic interrogation (at least in the NHP model).

I have been brought in as an additional reviewer and will focus the following paragraph on the response to reviewers the authors have carefully provided. In doing so, I would be doing the authors a disservice if I did not point out that the authors were given a difficult task. One of the reviewers did not carefully read the manuscript and negatively dismisses the manuscript as "a paper from a past era" potentially because of unfamiliarity with the complexity of the current debate about LIPs function. Much of the criticism reflects on aspects that are inferred by the reviewer and that are certainly not present in the current form of the manuscript. For example, the authors clearly demonstrate that LIP computes and accumulates evidence based on "arbitrary" sensory inputs which is a significant and important finding. It is also important to note however that the generalization of that aspect cannot be assessed in the current manuscript. Moreover, the two reviewer's insistence of manipulation for causality goes against the notion of the manuscript since single neuron specific alterations are impossible (at least as far as I am aware) in the NHP and certainly not within LIP that is buried within the IP sulcus. As far as I understand the manuscript the main claim is the local computations so perturbations would not yield any additional evidence to that argument. This notion is also present in a different critique by reviewer 3

(noted as major) about the strategy the animals employed. The authors have clearly delineated between the strategies by demonstrating that the neurons accumulated the difference between the reward probabilities.

All in all, I am a bit puzzled by what happened to this manuscript. This is a great paper with innovative behavior, solid neurophysiology and a clear goal that is achieved. One of my main concerns with the manuscript is the obvious difference between the two animals studied on multiple aspects of the analysis. It would be interesting to find out why these differences occurred but given the fact that the main effects replicate this should not make the manuscript any less impactful. Future studies using modern probes with higher channel counts will likely circumvent these shortcomings.

The authors have addressed all critiques by the reviewers, and I do not see anything major beyond my personal curiosity that needs to be addressed in this manuscript.

We would like to thank reviewers for their constructive comments. Here, we provide a point-by-point response. Below, the reviewers' comments are quoted in black; our replies are in blue.

REVIEWERS' COMMENTS

Reviewer #1 (Remarks to the Author):

I previously reviewed this manuscript for [another journal]. The authors have addressed all of my concerns from that previous review. However, there is one point I would like to address. In the Introduction (lines 55-57), the authors state "We reason that if LIP inherits evidence accumulation signal from somewhere else, we would unlikely observe computation stages that happen before the accumulation." One could argue that we may still observe such stages, but the underlying "accumulation" signal would just not be used for evidence accumulation and possibly for some other computation in the decision or motor process. It would greatly benefit the reader if the authors addressed this directly.

We agree with the reviewer that the accumulated evidence represented by the LIP neurons may be used for computations other than decision making. Yet, this is not the point that we are trying to make in the introduction. We are trying to argue that the 'accumulation' signal, no matter how it may be used in the brain, is computed locally in the LIP.

Reviewer #2 (Remarks to the Author):

Zhang and colleagues present a novel and exciting finding suggesting that the lateral intraparietal area locally accumulates information and computes evidence for decision making. This is a very important finding, especially in light of recent debates about distributed representations in frontal cortex that allow for little systematic interrogation (at least in the NHP model).

I have been brought in as an additional reviewer and will focus the following paragraph on the response to reviewers the authors have carefully provided. In doing so, I would be doing the authors a disservice if I did not point out that the authors were given a difficult task. One of the reviewers did not carefully read the manuscript and negatively dismisses the manuscript as "a paper from a past era" potentially because of unfamiliarity with the complexity of the current debate about LIPs function. Much of the criticism reflects on aspects that are inferred by the reviewer and that are certainly not present in the current form of the manuscript. For example, the authors clearly demonstrate that LIP computes and accumulates evidence based on "arbitrary" sensory inputs which is a significant and important finding. It is also important to note however that the generalization of that aspect cannot be assessed in the current manuscript. Moreover, the two reviewer's insistence of manipulation for causality goes against the notion of the manuscript since single neuron specific alterations are impossible (at least as

far as I am aware) in the NHP and certainly not within LIP that is buried within the IP sulcus. As far as I understand the manuscript the main claim is the local computations so perturbations would not yield any additional evidence to that argument. This notion is also present in a different critique by reviewer 3 (noted as major) about the strategy the animals employed. The authors have clearly delineated between the strategies by demonstrating that the neurons accumulated the difference between the reward probabilities.

All in all, I am a bit puzzled by what happened to this manuscript. This is a great paper with innovative behavior, solid neurophysiology and a clear goal that is achieved. One of my main concerns with the manuscript is the obvious difference between the two animals studied on multiple aspects of the analysis. It would be interesting to find out why these differences occurred but given the fact that the main effects replicate this should not make the manuscript any less impactful. Future studies using modern probes with higher channel counts will likely circumvent these shortcomings.

The authors have addressed all critiques by the reviewers, and I do not see anything major beyond my personal curiosity that needs to be addressed in this manuscript.

Jan Zimmermann

Thanks

We thank Dr. Zimmermann for the careful reading and constructive comments.

Regarding the difference between the two animals, this is indeed interesting but cannot be resolved with the current data. The different strategies adopted by the two monkeys can only be revealed in the neural data and cannot be distinguished behaviorally. As the reviewer correctly pointed out, further studies with a larger neural dataset will be needed.